# Social Media Sentiment about COVID-19 Vaccination Predicts Vaccine Acceptance among Peruvian Social Media Users the Next Day

**DOI:** 10.3390/vaccines11040817

**Published:** 2023-04-07

**Authors:** Ayse D. Lokmanoglu, Erik C. Nisbet, Matthew T. Osborne, Joseph Tien, Sam Malloy, Lourdes Cueva Chacón, Esteban Villa Turek, Rod Abhari

**Affiliations:** 1Department of Communication Studies, Northwestern University, Evanston, IL 60208, USA; 2Department of Mathematics, The Ohio State University, Columbus, OH 43210, USA; 3MITRE, McLean, VA 22102, USA; 4School of Journalism and Media Studies, San Diego State University, San Diego, CA 92182, USA

**Keywords:** COVID-19, vaccine acceptance, social media, sentiment analysis, social amplification of risk framework, Peru

## Abstract

Drawing upon theories of risk and decision making, we present a theoretical framework for how the emotional attributes of social media content influence risk behaviors. We apply our framework to understanding how COVID-19 vaccination Twitter posts influence acceptance of the vaccine in Peru, the country with the highest relative number of COVID-19 excess deaths. By employing computational methods, topic modeling, and vector autoregressive time series analysis, we show that the prominence of expressed emotions about COVID-19 vaccination in social media content is associated with the daily percentage of Peruvian social media survey respondents who are vaccine-accepting over 231 days. Our findings show that net (positive) sentiment and trust emotions expressed in tweets about COVID-19 are positively associated with vaccine acceptance among survey respondents one day after the post occurs. This study demonstrates that the emotional attributes of social media content, besides veracity or informational attributes, may influence vaccine acceptance for better or worse based on its valence.

## 1. Introduction

The relationship between online discourse, especially in social media, and vaccine hesitancy has garnered scrutiny by academics and policymakers for some time. However, this concern about social media and vaccine hesitancy has been exacerbated by the rapid proliferation of online information about COVID-19 across multiple social media platforms and its possible influence on COVID-19 vaccination [1,2,3,4]. Understanding how online discourse may influence attitudes about COVID-19 vaccination globally is important for vaccine promotion and mitigating current and future pandemics worldwide.

Unfortunately, the literature on the relationship between online discourse and vaccination attitudes and behaviors historically has focused on what are typically termed WEIRD populations—Western, Educated, Industrialized, Rich, and Democratic [5]. In a recent scoping review, for instance, of the 352 studies initially collected from Web of Science about vaccine hesitancy and online content published between 2000 and 2020, seventy-five percent of them (254) were located in North America or Western Europe [6]. More generally, other scholars have identified a similar bias in research about how media, including social media, amplify or dampen risk perceptions [7]. Commentators have also noted major gaps in our understanding of the effects of COVID-19 online information in languages other than English, such as Spanish [8]. Moreover, most examination of online discourse about vaccination, including COVID-19 vaccination, has been descriptive (e.g., [9,10]), without establishing a clear link with vaccine attitudes or behaviors ([6], although see [2,11,12] for exceptions).

This paper contributes to closing some of these gaps by directly examining the relationship between online discourse about COVID-19 vaccination and its acceptance in Peru, a non-WEIRD, middle-income, non-English language context that has suffered the greatest number of excess deaths from the COVID-19 pandemic, relative to their population, in the world [13]. We begin by presenting a theoretical outline of why online discourse may influence vaccine acceptance, drawing upon the Social Amplification of Risk Framework—otherwise known as SARF [14,15], and the important roles that affective [16] and trust [17] heuristics play in influencing risk decisions and behaviors. Based on this research, we hypothesize that the net (positive) sentiment and trust emotions expressed in social media posts about COVID-19 vaccination will forecast vaccine acceptance among social media users.

Our analyses include a dataset of nearly nine-million tweets from influential Peruvian online communities collected daily during the early stages of vaccination rollout in Peru between December 2020 and August 2021, and daily survey data collected within Peru during the same corresponding time period by the University of Maryland Global COVID-19 Trends and Impact Survey [18]. Employing computational methods and machine learning, we measure the daily topic prominence of COVID-19 vaccination and expressed emotions in the collected online discourse. We then use a time series analysis to test whether this sentiment about COVID-19 vaccination is associated with the daily percentage of Peruvian survey respondents accepting the COVID-19 vaccine. Our findings show that net (positive) sentiment and trust emotions expressed in tweets about COVID-19 are positively associated with vaccine acceptance among survey respondents one day after the post occurs.

### 1.1. Social Amplification of Risk and Social Media

Risk perceptions are subjective interpretations of the potential hazards, events, or behaviors in the face of uncertainty that derive from biased assessments of complex and ambiguous risk ‘signals’ found in one’s social surroundings [15,19]. The social amplification of risk framework (SARF) argues that this subjective processing of risk signals occurs in environments that combine individual and social dynamics with institutional, political, economic, and cultural factors [14]. In other words, the interplay between socially constructed risk signals and individual psychological processes determines the amount of perceived risk and the consequences for judgment and decision making, as is the case for vaccine acceptance.

The SARF literature shows that “social stations”, such as mediated communication channels (e.g., news, social media), play an important role in communicating risk signals to individuals and influencing risk perceptions, as not all risks are directly experienced by individuals (e.g., [14]). This influence of mediated channels has been found to be especially important when considering health risks, including vaccination (e.g., [20,21]). SARF asserts that mediated communication channels may over-emphasize or under-emphasize certain aspects of risk based on the volume or prominence of information, metaphors, images, symbols, emotions, and tone used in its content, thereby biasing the type of information an individual receives about risk [14,22]. The contribution of mediated channels such as social media to individuals’ risk interpretations may be substantial, as they not only provide information about risk traits, but also about the risk responses of others.

This later dimension of mediated risk signals is especially important when considering how social media discourse may influence risk behaviors. The thoughts, feelings, and behaviors of individuals cannot be fully comprehended without assessing their interactions in the real or imagined presence of other people [23] (see also [24,25,26]). Social interactions create socially derived frames of reference that help people make sense of the world around them, especially when dealing with ambiguity or complexity [27]. In this sense, social media platforms have become important “social stations” in which risk signals are created, diffused, and contested by a diverse range of individual and organizational actors, as well as attenuating or amplifying risk signals from other social stations such as news and official sources [7].

Applying SARF to understanding risk signals in social media and their consequences is still an emerging field of inquiry, and research outside of WEIRD contexts is even scanter [7]. In recent years, scholars have begun to apply SARF to examine how social media communicates risk signals about health and environmental topics [7,20,28,29,30], including COVID-19 (e.g., [31]). Most of these studies largely fall into two categories—those that primarily describe the content attributes (e.g., the prominence of topics, sentiment, emotions, targets) of social media content (e.g., [7,20]) and those that rely on survey data to examine how risk perceptions vary across different types of social media and traditional media channels or content (e.g., [32]).

However, unlike examinations of SARF in news media [33], to date, there have been few, if any, studies integrating the analysis of social amplification or attenuation of risk in social media content with a direct test of its impact on downstream risk perceptions, attitudes, or behavior. This study is one of the first to address this lacuna by examining the social media amplification of risk about COVID-19 vaccination, specifically the volume or prominence of COVID-19 vaccination as a topic in social media posts, the sentiment (positive vs. negative), and trust emotions expressed in them, and their downstream relationship with social media users’ vaccine acceptance.

### 1.2. Social Media Risk Amplification, Heuristics, and Vaccine Acceptance

SARF does not provide testable hypotheses on how the social amplification of risk in social media content influences risk perceptions, attitudes, and behavior. To understand the connection between social media content and risk attitudes and behaviors, we turn to the role of mental heuristics in influencing attitude formation and decision making, especially when making complex decisions with uncertainty or considering a new potential risk such as a new vaccine [15,34,35,36]. Through this theoretical lens, we may link the content attributes of social media posts about COVID-19 vaccination with specific cognitive heuristics that people use to assess risk and make decisions about vaccination.

For instance, the risk may be attenuated or amplified by the tone and emotions associated with the object of risk in (social) media content [7,20,22,28,37]. This is especially relevant when considering social media content as the medium that promotes the sharing and distribution of highly emotional news and posts, with the prevalence of emotional content rising in response to social, political, economic, and health events [38,39].

Therefore, the emotional valence, positive versus negative, associated with a risk in social media may influence risk perceptions and decision making through the affective heuristic [16,34,40,41]. The affective heuristic is a common mental shortcut used to evaluate the risks and benefits associated with a behavior or object based on a person’s overall feelings, or net sentiment, about it. Net positive feelings or sentiments are associated with greater perceived benefits and lower perceived risks. Conversely, net negative feelings or sentiments are associated with greater perceived risks and lower perceived benefits [16,34]. This affective heuristic is a key determinant of vaccine acceptance [34,42,43], including the COVID-19 vaccine [44]. Positive and negative sentiments associated with a behavior or risk in social media content—such as vaccination, for example—may therefore be employed as a mental shortcut that influences how social media users feel about risk perceptions and the likelihood of engaging in the behavior. Thus, we posit that the net (positive minus negative) sentiment expressed in social media posts about COVID-19 vaccination will be associated with greater vaccine acceptance among social media users (H1).

Beyond the overall positive and negative sentiment or feelings about COVID-19 vaccination, another key emotion that may be expressed in social media content and is essential for vaccine acceptance is the discrete emotion of trust [1,45,46]. Trust is an emotional heuristic that may be employed as a mental shortcut by people to assess risk and guide behavior when making decisions under uncertainty with greater trust associated with lower risk—especially with new or unfamiliar risks [4,17,47]. Moreover, accurate online information is more likely to elicit this emotion in social media posts [48]. Consequently, we would expect that social media posts about COVID-19 vaccination that express a higher level of trust would be associated with lower perceived risk and, thus, greater vaccine acceptance among social media users (H2).

## 2. Materials and Methods

### 2.1. Study Context: Peru

Our examination of the social media amplification of risk and its downstream impact on risk decision making is situated within the study context of Peru. Peru represents an important national context for examining how social media risk amplification may influence COVID-19 vaccine acceptance based on several key factors. First, as a non-Western, developing, non-English speaking country, it addresses historical WEIRD biases and gaps in the social amplification of risk [7], and social media and vaccine acceptance [6] research.

Second, in terms of COVID-19 specifically, Peru has the highest P-score (a comparative cross-national measure that considers differences in population size) of excess deaths in the world due to the COVID-19 pandemic [13,49]. Peru also has the highest level of baseline vaccine hesitancy in Central and South America, with 64% of Peruvians agreeing that vaccines are generally safe compared to the regional and national average of 76% [50].

Third, from a social media perspective, in the early days of the COVID-19 pandemic, the Peruvian online environment was heavily penetrated by misinformation [51,52,53,54]. Although the official Twitter account of the Peruvian Ministry of Justice and Human Rights announced in April 2020 that spreading misinformation on COVID-19 would be punishable by prison [52,55], this did not stop the spread of mis- and disinformation in Peru. Most of the information campaigns came from different groups in civil society and politicians—including candidates for congress and presidential candidates—and some right-leaning media were involved in most cases that spread mis- and disinformation regarding the effectiveness of COVID-19 vaccines [53,55,56,57,58,58,59]. According to some experts, the disinformation campaign might have caused the vaccination rollout delay [60,61].

### 2.2. Methodology

#### 2.2.1. Step 1: Data Collection—Sentinel Surveillance

Twitter is a very large online landscape, with the hashtag #COVID-19 having been tweeted 628,809,016 times between 1 January and 9 May 2020 [62]; thus, tracking every COVID-19 post on Twitter is near impossible (given API rate limits). However, conversations on Twitter are often driven by a small set of “elites” (small, relative to overall Twitter users), and this appears to be the case for the discussion surrounding COVID-19 [63]. Monitoring Twitter for COVID-19 conversations is thus analogous to the public health technique known as *sentinel surveillance*, a methodology introduced by Osborne et al. [64]. In public health, sentinel surveillance systems collect data from a limited number of reporting sites, otherwise referred to as sentinels. Such an approach is desirable when it is not feasible to perform comprehensive surveillance [65]. In the setting of widespread unreliable online information about COVID-19, sentinel surveillance is appropriate because it offers a method for distilling the entire COVID-19 Twitter conversation in a geographic region down to the posts of a few hundred or thousand sentinel node accounts [64].

A key aspect of our approach is our use of network science tools to guide sentinel node selection. Twitter nodes are naturally linked to one another through relationships such as retweets, links to media, and other websites. These relationships can be used to form different types of networks whose structure can be analyzed using tools from network science [66]. These include the identification of tightly knit groupings of Twitter accounts that form ‘communities’ and the identification of ‘central’ nodes that are influential in shaping the regional Twitter conversation.

A two-step approach, following the methodology of Osborne et al. [64], was used for sentinel node identification in the context of COVID-19 discourse on Peruvian Twitter: (1) an initial phrase search using the Twitter API for COVID-19-related terms, followed by community detection to identify initial ‘seed’ accounts of interest for the Peruvian Twitter ecosystem, and (2) the identification of accounts that were heavily retweeted by the initial seed accounts regarding COVID-19, using snowball sampling. The snowball sampling method implies that they are significant users of the initial set. The most highly retweeted nodes from this second retweet network were combined with our initial seed set to form the final sentinel set. This approach is analogous to respondent-driven sampling (RDS), a widely used approach in epidemiology for sampling from individuals in difficult-to-reach segments of the population [67].

Two different phrase searches were conducted to narrow the final sentinel set to COVID-19 topics in the Peruvian Twittersphere: non-geolocated search for Spanish-language tweets in the sentinel set containing the phrases ’COVID’ or ‘vacuna’ and tweets containing the phrase ‘COVID’ that were geolocated within one thousand miles of Lima. Non-geolocated Twitter queries search through all tweets posted in a recent time window containing the search terms, while geolocated searches only search tweets posted by users who have enabled Twitter to know the location of the device used to post their tweets. Studies have reported that as few as 0.85% [68] of tweets may be geolocated, raising the possibility of selection bias, which is why we adopted this dual search approach to the final sentinel set. Since the tweets were from the final sentinel set, we are as confident as Twitter API on their location.

Non-geolocated Spanish language searches in the sentinel set for ‘COVID’ or ‘vacuna’ yielded 288,180 tweets sent by 171,835 unique nodes. A weighted, directed retweet network was constructed from these data, with nodes corresponding to Twitter accounts and the weight of the arc from *j* to *i* corresponding to the number of times that account *j* retweeted node *i*. Community detection was then performed on the largest connected component of this retweet network to identify tightly knit groupings of nodes. Specifically, we used modularity maximization [66,69] via a Louvain method [70], adapting code from [71,72] relating to directed graphs (see Equation (1) in [71]. Community detection yielded a cluster of 1889 nodes that were relevant for Peru, which were labeled as non-geolocated seed nodes upon inspection.

Searches for tweets containing ‘COVID’ geolocated in the sentinel set within one thousand miles of Lima yielded 20,969 tweets sent by 13,263 unique accounts. Community detection on the retweet network for the geolocated data yielded thirty-six communities. The resulting communities were inspected, and communities not obviously relevant to the Peruvian online ecosystem were discarded. The remaining 130 nodes were retained as geolocated seed nodes. Combining the non-geolocated and geolocated seed nodes gives our combined seed node set.

Community detection was performed to identify groupings of accounts. We first identified communities in our combined seed node set to expand the coverage breadth and identify influential Twitter accounts to include in our final sentinel node set. We then identified highly retweeted accounts from each community to include in our sentinel nodes. The fifty most highly retweeted nodes about COVID-19 from each detected community were then added to the combined seed nodes to give an initial sentinel node set. The most recent 3200 tweets were obtained from each node in our combined seed node-set, and a retweet network was constructed from tweets containing any of the following keywords related to COVID-19: ‘COVID’, ‘coronavirus’, ‘pandemia’, and ‘vacuna’.

The eighty most highly retweeted nodes in each community were identified as our final 739 sentinel nodes to be tracked longitudinally. From this step, we treated the community labels to be permanently associated with each sentinel node. Once the sentinel node accounts were determined, the premium Twitter API was used to collect 231 days of tweets from each user to analyze from 21 December 2020 to 8 August 2021. A total of 8,725,573 million tweets were collected for all the sentinel communities, creating the corpus for our semantic analysis [64]. We started our data collection in December 2020 because the vaccination rollout in Latin America began in December 2020 with Mexico, Chile, Argentina, and Costa Rica [73,74]. Peru managed to vaccinate 25% of the population in August 2021, when we concluded our data collection, following the methodology of Pierri et al. [2,75]. The social media data collected from Twitter are representative of the online-eco system of Peru for COVID-19.

#### 2.2.2. Step 2: Measuring Social Media Amplification of Risk

Previous research examining epidemics has utilized unsupervised machine learning and automated content analysis as inductive approaches to examining large copra and minimizing researcher bias [76]. Similarly, research examining social media posts employed unsupervised machine learning and semantic analysis to categorize the semantic map of social media conversations [1,77,78,79]. Topic modeling, or Latent Dirichlet Allocation (LDA), is a computational content analysis tool surfacing the “hidden thematic structure of a collection of text” [80]. The LDA algorithm models the representation of the words, with each other, within a document and the corpus, through “topics” [80]. These facilitate researchers to label topics inductively by using both the words within each topic and the documents in each topic. Thus, LDA analysis allows a document to represent multiple topics, providing a deeper insight into the thematic structure of the corpus. All the following steps were carried out using R version 4.2.2 through the editor R Studio 2022.12.0.

The data were preprocessed following previous methods for conducting unsupervised machine learning [76,81]. The language composition of the corpus is 78.41% Spanish, 18.07% English, and 3.52% other languages. Thus, for the rest of the preprocessing and analysis, we utilized both Spanish and English in the computational process. After the preprocessing, we visualized the optimal number of topics using the four metrics: Grifffiths 2004, Cao Juan 2009, Arun 2010, and Deveaud 2014 [82]. We concluded that the highest quality results were returned from sixty-five topics with an alpha of 0.1 (see Appendix A).

The identified thematic associational clusters included the top unique words per topic (see Appendix A) and the documents that best illustrated the presence of each topic [81]. Employing the top words and tweets by four English-speaking coders—two native Spanish speakers (Peruvian and Colombian) and two proficient Spanish speakers—we inductively collapsed the fifty-seven topics (8 topics were boiler-plate topics) into twelve topic clusters to later model how they are associated with attitudes and behavior around vaccination. We assigned labels to each topic (Figure 1) and selected the topics on COVID-19 vaccination.

Based on our 65-topic model, COVID-19-related topics included 41.2% of the Peruvian dataset across different subtopic categories. Among the COVID-19-related topics, COVID vaccination was clearly the most prominent, including 8.2% of the tweets, with the most temporal variability and the greatest standard deviation of prevalence (SD = 0.7%) (for details on the other clusters, please see Appendix A). Thus, to produce cluster prominence, we continued our modeling with the COVID-19 vaccination topic cluster and aggregated the gamma scores of tweets per day.

Sentiment analysis is an unsupervised dictionary method where lexicons code words for sentiments and emotions [83]. Studies show the relationship between overall sentiments on Twitter as predictors of behavior and attitudes regarding vaccination [1,84,85,86,87,88]. In addition to the topical content, the sentiment of the tweets provides a deeper understanding of the nature of the conversation in our topic clusters and how these are associated with downstream effects. We used the NRC Spanish and English lexicons to calculate the sentiment of the words within each tweet (see Appendix A) [89]. The NRC lexicon was previously used to examine Twitter data sentiments and conversations around COVID-19 and vaccinations [79,86]. The package provides a score for each sentiment category, including trust, and an overall net sentiment score; the dictionary was established and verified with crowdsourcing [89]. Following the literature on semantic analysis and social media conversations about vaccination, we opted to examine trust as an emotion [17,90] and overall net sentiment (negative sentiment subtracted from positive sentiment) in our analysis. The lexicon approach does not consider context, such as the sentence could be satirical, cynical, or using a word with local or temporal homonyms, and the word-sentiment scoring does not consider preceding and anterior words in sentiment calculations [89]. To address these two limitations of the lexicon approach, our study examines the association between the net sentiment and trust of the text, and not the sentiment towards the content. We multiplied each sentiment score by the tweet’s topic probability to calculate the weighted topic sentiment score as our first and second independent variables, the net sentiment, and the trust scores per day of the topic cluster of COVID-19 vaccination (see equation below and Appendix A for an example of sentiment calculations).
Net Sentiment of Topic Cluster COVID Vaccination=Net Sentiment Score of Tweet×Topic Probability
Trust Emotion of Topic Cluster COVID Vaccination=Trust Score of Tweet×Topic Probability

#### 2.2.3. Step 3: Assessing Downstream Effects of Social Media Amplification of Risk on Vaccine Acceptance

Employing our measures of the social media amplification of risk about COVID-19 vaccination (topic prominence and sentiment analysis), we tested our hypotheses on how social media amplification is associated with COVID-19 vaccine acceptance among Peruvian social media users by using survey data from the UMD Social Data Science Center Global COVID-19 Trends and Impact Survey [18,91,92] and a time series analysis. At the start of 2022, around 83% of Peru’s total population were social media users, and 74.4% were Facebook users [93]. The high penetration of Facebook in Peru creates confidence in using the survey results to test associations with social media content. Furthermore, previous studies demonstrated the importance of testing offline attitudes and behaviors with the thematic analysis of online content in public health crises such as COVID-19, in order to show possible venues for effective public health campaigns and interventions [76,94].

The UMD Global COVID-19 Trends and Impact Survey is a daily tracking survey of Facebook users conducted in 115 countries, including Peru [95,96,97,98]. The survey is conducted through Facebook recruitment that is weighted to represent the characteristics of each national population, with non-response and missing data treated as Missing Completely at Random (MACR) and computed with non-response weights with an inverse propensity score weighting (IPSW) approach (for details on survey methodology, please refer to [91,92,99]). The daily aggregated population estimates for all survey questions are publicly available through the project’s open data API [18] (see Appendix A for sample sizes). Detailed information on the dependent and independent variables used in the analysis are in Table 1 and Table 2. Since the vaccination rates in Peru do not plateau until early 2022, we measured vaccine hesitancy with the same methodology as Pierri et al. [2].

We conducted Vector Autoregressive Models (VAR) using the daily theme probability and sentiments of the theme as independent variables, and controlled for linear trends in the dependent variable, to test our hypotheses on the relationships between net sentiment (H1) and trust (H2) in social media posts about COVID-19 vaccination and vaccine acceptance. VAR multivariate time series models are the preferred time series methods when using topic prominence scores, as they allow for the effects of lags of each variable in the analysis [76,100,101]. With 231 continuous matching days of social media and vaccine acceptance data, we have sufficient units of analysis for a time series analysis [102].

First, we augmented our findings using a Dickey–Fuller test (ADF) to ensure the data were stationary and determine the optimal number of lags for the VAR models, as four days in sentiment prominence models [76,100]. Afterwards, we examined each independent variable as an individual model to eliminate multicollinearity. We used Impulse Response Functions (IRFs) to examine the effect size of the variables, as IRF allows us to trace a single shock in the systems of equations [103]. IRF demonstrates the evolution of the dependent variable in a given time frame after the impulse shock, which is the independent variable [104]. The time unit for IRFs corresponds to lags—in our case, days.

For a robustness check of reverse causality, we ran VAR using vaccine acceptance as the independent variable and net sentiment and trust in social media posts about COVID-19 vaccination as dependent variables (see Appendix A). We also ran our dependent variables with two other survey indicators to test the effects of vaccination and vaccination acceptance with family recommendations (see Appendix A).

## 3. Results

### 3.1. Risk Signal

The graph of the timeline of vaccine acceptance, trust, and net sentiment marked is presented in Figure 2.

### 3.2. H1 and H2: Social Media Amplification and Vaccine Acceptance

The model for affect heuristic predicting vaccination acceptance supports H1. We see that with an increase in net sentiment, there is an increase in vaccination acceptance (β = 0.58, *p* = 0.0102) (Table 3). Affective heuristic at lag one is associated with an increase in vaccination acceptance, with one standard deviation change in the net sentiment of the COVID-19 vaccination topic cluster predicting a 0.0029% [95% C.I. 0.00081–0.0048] change in vaccination acceptance (see Figure 3 for IRF), supporting H1.

The model for trust heuristic predicting vaccination acceptance supports H2. We see that a unit increase in trust increases vaccination acceptance (β = 0.55, *p* = 0.0150) (Table 3), with statistical significance at lag 1 (Figure 3), supporting H2. Trust emotion at lag one is associated with an increase in vaccination acceptance, with one standard deviation change in the trust emotion expressed in the COVID-19 vaccination topic cluster predicting a 0.0027% [95% C.I. 0.00078–0.0044] change in vaccination acceptance (Figure 3 for IRF), supporting H2. The reverse time series models for the robustness check had null results (Appendix A).

## 4. Discussion and Conclusions

Before discussing our results, we acknowledge some limitations of our study. First, we compared Twitter conversations to survey results collected from Facebook users in this paper. We are aware of the limitations of using two different platforms in our analysis; however, as Yang et al. found in their comparative study of online COVID-19 posts on Twitter and Facebook, there is a high degree of information overlap across platforms, especially from popular accounts [105]. Furthermore, Ginossar et al. established the cross-platform sharing of anti-vaccination content between YouTube and Twitter, emphasizing the normalcy of content overlap across different social media platforms [106]. Since our Twitter data were collected using sentinel surveillance, we argue that the semantic and sentiment analyses of COVID-19 vaccination content on Twitter are representative of content that circulates on Facebook and the online ecosystem more broadly.

Another caveat is that we only examined the conversations in the sentinel nodes. Using the sentinel nodes, we could home in on information spreaders, thus collecting data with a high likelihood of sharing across platforms. Thus, the research was limited to social media users. In addition, as described above, examining all the Peruvian conversations regarding COVID-19 vaccination on Twitter over the nine-month period was not the best choice in terms of size and feasibility [64].

Our sentiment analysis also demonstrated a limitation. As the language of social media is dynamic and emojis are often used instead of words, we would like to further interrogate the effect of emojis on sentiment measures. Future studies should test with and without emojis in the sentiment analysis to observe the differences and draw conclusions. Finally, although the effect sizes of net sentiment and trust in social media posts about COVID-19 vaccination on vaccination acceptance were small, the overall magnitude of their effects on attitudes may be substantive in the long term when taking into account the total influence of daily, repeated, cumulative exposure to such mediated content [107].

Despite the limitations, this study expands our understanding of how social media content may influence COVID-19 vaccine acceptance and the role that social media plays in amplifying risk. Our study presented both a theoretical framework for linking the social amplification of risk in social media content with risk perceptions and behaviors, while simultaneously demonstrating a methodological pathway to test it. While prior research has implied explicit linkages between social media content and risk behaviors, our study is one of the first to directly connect the content emanating from the social media amplification of risk with population risk behaviors in the context of SARF. This study helps address the pressing problem of a lack of SARF, social media, and vaccine acceptance research in non-English, non-WEIRD contexts and languages.

The analyses were consistent with our hypotheses on how the social media amplification of risk may influence risk behaviors through common heuristic biases people employ in order to make sense of risk situations and make decisions with uncertainty. The presidential elections and political candidates using COVID-19 vaccines within their campaigns helped establish the risk signals. The change in net sentiment and trust corresponding to these dates is consistent with the research to date, as key political figures’ information and attention to vaccination changed the behavioral attributes within the social media discourse. For example, several major Peruvian political figures promoted arguments discrediting COVID-19 vaccines by claiming that vaccines were used to develop animal genes in people [59] and that there was no proof of their effectiveness and safety [58,108], that they caused severe side effects in certain populations [57], and that the Sinopharm vaccine caused more infections than the placebo used in the tests [56].

Our results were also consistent with how the affective heuristic shapes risk behaviors. The analysis showed how the net sentiment (positive sentiment minus negative sentiment) in social media posts about COVID-19 vaccination was associated with a change in vaccine acceptance the day after the post. Our interpretation is that when the net (positive) sentiment about COVID-19 vaccination increased in social media discourse, there was a corresponding increase in vaccine acceptance the next day. Conversely, vaccine acceptance declined if net sentiment decreased (i.e., it became more negative). This co-variance of vaccine acceptance with the valence of the net affect expressed in social media content is consistent with how the affective heuristic influences risk perceptions and decision making.

Similarly, the relationship between the trust heuristic and COVID-19 vaccination acceptance illustrated a positive effect one day after an increase in trust emotion in the social media content. These results further corroborate the research [1,4,45,46], where trust is a mental shortcut for lowered risk perception. Thus, our results demonstrate a clear link between how social media conversations’ sentiments and emotions influence risk behaviors.

The methodology utilized in the paper conveys a generalizable model for future studies examining the social media amplification of risk in social media posts, including the content of online news or videos found on social media and its downstream influence on risk perceptions and behavior across a range of health and environmental risk contexts. By employing a dual language topic modeling and sentiment analysis, we could account for the varied nature of social media and quantify the conversations for further testing. This is especially crucial in expanding and comparing research across different countries. Furthermore, by using social media conversations, we were able to test the informal networks of information and expand beyond traditional media sources in our analysis.

Our study has both theoretical and practical implications for understanding the relationship between risk communication, social media content, and vaccine acceptance. Our data collection and analysis of social media content included all vaccine-related content, regardless of accuracy or credibility. This differs from other studies that only focus on false or misleading content and suggests the primacy of emotional valence over the credibility or accuracy of content when considering its association with risk perceptions and behavior. Moreover, from an information processing perspective, we know that affective responses to content are processed more quickly than cognitive responses—and in fact, often color cognitive outcomes. Thus, our theoretical contribution sheds light on how emotional cues embedded within social media content may influence risk attitudes and behaviors beyond any informational attributes, such as veracity.

For public health communicators, our study demonstrates that social media content may be a double-edged sword for vaccine promotion; it is associated with the amplification or dampening of vaccine acceptance depending on the emotional valence of its content. The findings provide guidance for public health communicators and others engaged in vaccine promotion by reinforcing the need to employ language that communicates positive emotions and trust when communicating about COVID-19 vaccination—and may be a vital component of messaging, even more so than credibility.

In sum, the main goal of our study was to examine if the social media amplification of risk about COVID-19 vaccination through emotional cues was associated with vaccination acceptance in social media users. Ultimately, we found that it was, and the effect was observed only a day later. This is important for future work on vaccination acceptance, as it suggests a means for increasing vaccination acceptance through the strategic use of emotional cues in online content about vaccination.

## Figures and Tables

**Figure 1 vaccines-11-00817-f001:**
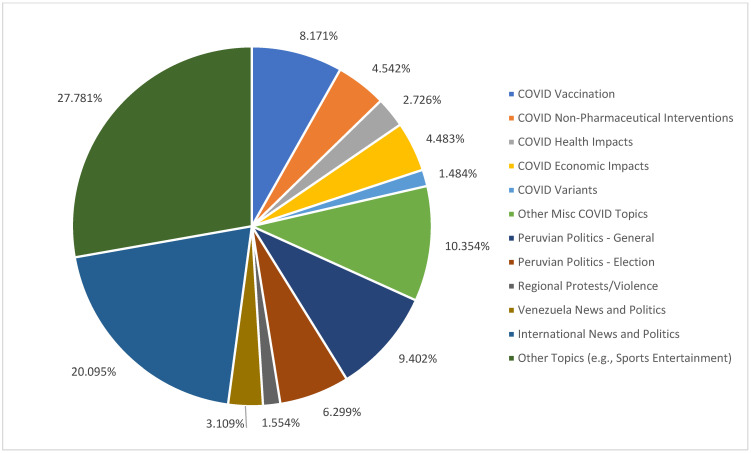
Topic clusters distribution.

**Figure 2 vaccines-11-00817-f002:**
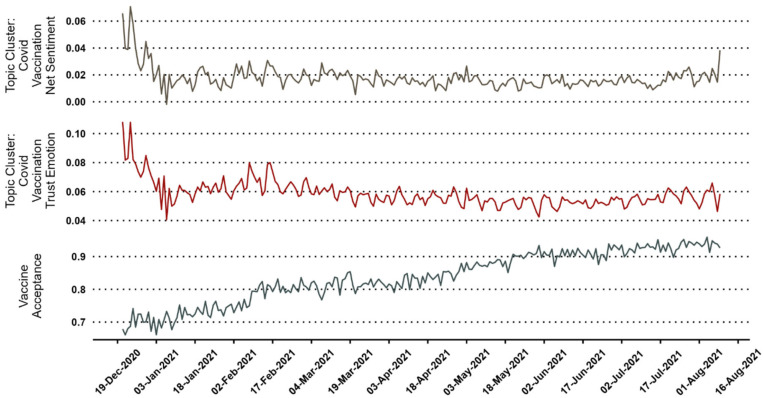
Trust, net sentiment, and vaccine acceptance from December 2020 to August 2021.

**Figure 3 vaccines-11-00817-f003:**
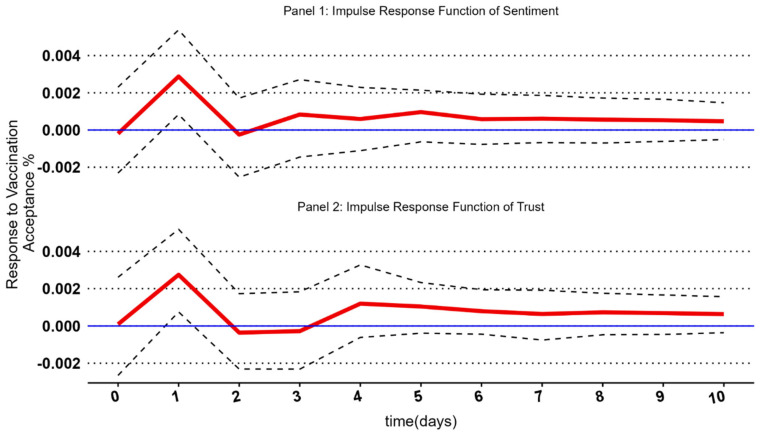
The impulse response function of two sentiment measurements of the COVID-19 vaccination cluster on the dependent variables. The IRF illustrates the response size of each shock to the system. The red line represents each lag’s main IRF point estimates. These can be interpreted as the effect of one standard deviation change.

**Table 1 vaccines-11-00817-t001:** Variables and definitions.

*Variable Name*	*Definition*
*Independent Variables (x)*	
Topic Cluster: COVID-19 Vaccination Sentiment	Topic prominence per tweet multiplied by the net sentiment (positive–negative sentiment count) calculated by the NRC lexicon
Topic Cluster: COVID-19 Vaccination Trust Emotion	Topic prominence per tweet multiplied by the trust emotion count calculated by the NRC lexicon
*Dependent Variables (y)*	
Vaccination Acceptance	Respondents who had a vaccine, an appointment to get vaccinated or who would definitely or probably choose to get vaccinated if a COVID-19 vaccine were offered to them, a value between 0 and 1, from The UMD Social Data Science Center Global COVID-19 Trends and Impact Survey

**Table 2 vaccines-11-00817-t002:** Descriptive statistics of variables.

Statistic	N	Mean	St. Dev.	Min	Pctl (25)	Pctl (75)	Max
Topic Cluster: COVID-19 Vaccination Sentiment	231	0.018	0.008	−0.002	0.013	0.020	0.071
Topic Cluster: COVID-19 Vaccination Trust Emotion	231	0.059	0.009	0.041	0.053	0.061	0.108
Vaccination Acceptance	230	0.84	0.077	0.661	0.791	0.905	0.959

**Table 3 vaccines-11-00817-t003:** VAR models of net sentiment and trust emotion of COVID-19 vaccination topic cluster predicting vaccination acceptance.

MODEL 1	Dependent Vaccination	Variable Acceptance	(y):	MODEL 2	Dependent Vaccination	Variable Acceptance	(y):
	Coefficients	Confidence Interval			Coefficients	Confidence Interval	
Sentiment (lag 1)	0.58 * (0.22)	[0.1385, 1.0184]		Trust (lag 1)	0.55 * (0.23)	[0.1084, 0.9988]	
Sentiment (lag 2)	−0.38 (0.24)	[−0.8575, 0.1001]		Trust (lag 2)	−0.41 (0.25)	[−0.9078, 0.0788]	
Sentiment (lag 3)	0.02 (0.24)	[−0.4587, 0.4971]		Trust (lag 3)	−0.18 (0.25)	[−0.6754, 0.3034]	
Sentiment (lag 4)	0.03 (0.21)	[−0.3746, 0.4410]		Trust (lag 4)	0.32 (0.21)	[−0.0975, 0.7392]	
const	0.18 ** (0.05)	[0.0727, 0.2838]		const	0.17 ** (0.06)	[0.0614, 0.2797]	
trend	0.0003 ** (0.00)	[0.0001, 0.0004]		trend	0.0003 *** (0.00)	[0.0001, 0.0005]	
Observations	227			Observations	227		
R^2^/R^2^ adjusted	0.943/0.941			R^2^/R^2^ adjusted	0.944/0.942		
log-Likelihood	592.96			log-Likelihood	593.66		

Notes: * Standard errors follow each model in parenthesis and confidence intervals. ** The first four lags were reported for each model with no instantaneous causality. *** Models control trends in the data.

## Data Availability

Study materials, code, and collected data of public accounts are available at https://osf.io/583pb/?view_only=c26689735c4e47a9845653ffecf0adfe (accessed on 23 March 2023) and https://github.com/nwccpp/peru-sarf (accessed on 23 March 2023).

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
