# Peer review of "Social Media Sentiment about COVID-19 Vaccination Predicts Vaccine Acceptance among Peruvian Social Media Users the Next Day"

_vaccines, 2023, doi:10.3390/vaccines11040817_

Round 1

Reviewer 1 Report

The Authors evaluated how COVID-19 vaccination Twitter posts influence its acceptance in Peru. The manuscript is clear and presented in a well-structured manner. The cited references are useful and relevant. The experimental design is suitable for testing the hypothesis but, as correctly highlighted by the Authors, the studied phenomena is complex and would require even more refined experimental design in future (e.g. emoji evaluation). The sample included in the study was correctly evaluated based on hypotheses to test. The manuscript’s results could be reproducible based on the details given in the Methods and Supplementary material section. The figures and tables are appropriate. The conclusions are consistent with the evidence presented if limitations are taken into account. Conflict of interest and funding statements are adequate.

Author Response

Dear Reviewer,

Thank you very much for the helpful and encouraging comments.

Reviewer 2 Report

Using Twitter messages to determine the positive and negative impact on acceptance of the COVID19 vaccine is an interesting study.

In general the study is well-executed computationally, although there are parts that need more clarity.

For example, given that the geolocating is poor, how do the authors ensure that the collected information was not from other spanish speaking areas? This could severely impact the outcome and prediction especially since Peru was the point and title of the research.

The other thing that needs a little more elaboration is that in general tweets positivity and negativity is difficult to tell apart. For example there could be a tweet about having fever after the vaccination and while at first glance it is a negative point, it also means that the user accepted the vaccination. Whether something is positive and negative may not necessary correlate easily under sentiment analysis. Perhaps provide some examples.

What would be helpful would be to somewhat describe the Twitter userbase in Peru. In some countries, Twitter may be used more by certain age groups or be less popular, etc. So this would lay the foundation.

I may have missed it, but I was not able to see any clear evaluation of vaccine acceptance survey done to determine how the correlation could be made between the Twitter responses and actual population acceptance.

It should be noted that social media do not often depict the real situation or real sentiment, and can be biased by the vocal few.

Some areas of English issues that may be corrected through more proof-reading.

All of the above should be addressed where possible, and if not, discussed.

Author Response

Dear Reviewer,

Thank you very much for the helpful comments. Below is the list of changes to the manuscript as advised:

  • The manuscript was edited as suggested, specifically:
    • We have added clarifications to the computational methodology (on page 7), and included an example of the sentiment and trust calculations in the online supplement (on pages 5 and 6 of the online supplement)
    • We have clarified the language on the data collection (on pages 4 and 5). The sentinel seeds collected were all geo-tagged. The sentinel node network that was then reproduced was through using snowball sampling. The snowball sampling method implies that they are significant users to the initial set, which we are reasonably confident reside in Peru. The most highly retweeted nodes from this second retweet network were then combined with our initial seed set to form the final sentinel set. The second step of the data collection, was the phrase search from the final sentinel set, to narrow the context to Covid-19 and vaccinations. In this step, the non-geo tagged tweets were from the snowball sampling in the first step.
    • Our sentiment analysis measures the sentiment within the text of the tweet and not the sentiment of the content. We made sure to make that clear when discussing sentiment analysis, and expanded on the limitations of sentiment calculations (on pages 6 and 7). Also in the Supplement S2 and S3 we provided an example of the calculation of the sentiment with the topic probability. 
    • We have included social media penetration in Peru to further demonstrate the high percentage of social media users in Peru (on page 7). We have added in the limitations that this study reflects only social media users (on page 11).
    • In the discussion we have included the limitations of the study being limited to social media users and limitations of the data collection method, acknowledging the possible bias of our findings (page 12).
    • We had a native English speaker proofread the manuscript.

Thank you very much again for the opportunity to revise our manuscript and we look forward to the reviews.

Reviewer 3 Report

The paper deals with an interesting topic. The reviewer has several concerns as described below. Hopefully authors could improve the paper. 

1. Abstract. The findings of the study need clearer explanations and methods should be described (sample, modeling methods). 

2. Introduction. sentence" A re-46 cent scoping review, for instance, of one hundred studies about vaccine hesitancy and 47 online content published between 2000 and 2020 found a substantial geographic bias fa-48 voring populations in high-income Western countries [6]. " should include what is the bias found. This is very relevant to the current study goals. 

3. Literature. "Social Media Risk Amplification " needs further expansion. What theories were used to support this and how this was used in the current study design. Is there any hypothesis proposed before data analysis?

4. Method. The data used two social media platforms, twitter and facebook. The reviewer has concerns whether these two platforms have the same audience pool. How to use twitter sentiment scores to measure the vaccination behaviors of Facebook users? This connection should be strongly argued to support this approach. In addition, the data collection period was December 2020- August 2021. The authors need a strong justification for the data collection periods. Why the particular time range was selected for analysis? 

5. Analysis. The authors seems to neglect the impacts of social relationships on the vaccination behaviors, such as family, friends, colleagues, etc. The authors are suggested to consider these important factors by including them as control variables in the analysis. 

6. Implications. The theoretical implications based on the study findings are missing. Please add. 

Best of luck with your research! 

Author Response

Dear Reviewer,

Thank you very much for the helpful comments. Below is the list of changes to the manuscript as advised:

  • The manuscript was edited as suggested, specifically:
    1. We have included details on sampling and methods to the abstract (page 1)
    2. We have included the findings of from the scoping review to bolster our argument (page 2)
    3. We have expanded the theoretical section on SARF, including how it was used in the current study (pages 2, 3 and 4). This study was not preregistered, but we did formulate hypotheses based on our theoretical framework which we tested with the collected data and analysis.
    4. We have included the explanation for the selection of the data collection dates at the end of the 2.2.1. Step 1: Data Collection – Sentinel Surveillance (on page 5). In this research, we employ the latent content (topic and sentiment) of Peruvian Twitter conversations about vaccination to represent the Peruvian social media information environment more generally. We have included citations 83, 84, 85 and 86 that utilize the same survey research outside of USA to bolster the reasoning for our research design (on page 7). We have also included social media penetration in Peru to further demonstrate the high percentage of social media users in Peru (on page 7). In the discussion section we added citation 94, a study that demonstrates the overlaps of social media content and users across different platforms (on pages 10 and 11).
    5. Since this is an aggregated time series analysis, the data and methodological approach cannot control for the impacts of social relationships on the vaccination behaviours. In our online supplement, we included a new model in table S7 that includes the covariate “Respondents more likely to get vaccinated if recommended by friends and family”, from the same UMD survey. The new model shows there is no significant association with Vaccination Acceptance with Family Recommendation with the dependent variable Vaccination Acceptance and the results supported our previous findings (online supplement page 9).
    6. We understand if our previous discussion section did not clearly delineate between the theoretical and practical implications of our study. We revised part of it to more clearly communicate the implications for both theory and practice. (page 13).

Thank you very much again for the opportunity to revise our manuscript and we look forward to the reviews.

Round 2

Reviewer 2 Report

The authors have addressed the comments in general. 

Some slight typos here and there, that can be corrected during the proofing. 

Reviewer 3 Report

Thank you for the successful revision. I don't have further comments. Congratulations!